# Transcriptomic Profile of Genes Regulating the Structural Organization of Porcine Atrial Cardiomyocytes during Primary In Vitro Culture

**DOI:** 10.3390/genes13071205

**Published:** 2022-07-05

**Authors:** Mariusz J. Nawrocki, Karol Jopek, Mariusz Kaczmarek, Maciej Zdun, Paul Mozdziak, Marek Jemielity, Bartłomiej Perek, Dorota Bukowska, Bartosz Kempisty

**Affiliations:** 1Department of Anatomy, Poznan University of Medical Sciences, 60-781 Poznan, Poland; mjnawrocki@ump.edu.pl; 2Department of Histology and Embryology, Poznan University of Medical Sciences, 60-781 Poznan, Poland; karoljopek@ump.edu.pl; 3Department of Cancer Immunology, Chair of Medical Biotechnology, Poznan University of Medical Sciences, 61-866 Poznan, Poland; markacz@ump.edu.pl; 4Gene Therapy Laboratory, Department of Cancer Diagnostics and Immunology, Greater Poland Cancer Centre, 61-866 Poznan, Poland; 5Department of Basic and Preclinical Sciences, Institute of Veterinary Medicine, Nicolaus Copernicus University in Toruń, 87-100 Toruń, Poland; maciejzdun@umk.pl; 6Physiology Graduate Program, North Carolina State University, Raleigh, NC 27695, USA; pemozdzi@ncsu.edu; 7Prestage Department of Poultry Science, North Carolina State University, Raleigh, NC 27695, USA; 8Department of Cardiac Surgery and Transplantology, Poznan University of Medical Sciences, 61-848 Poznan, Poland; kardiock@ump.edu.pl (M.J.); bperek@ump.edu.pl (B.P.); 9Department of Diagnostics and Clinical Sciences, Institute of Veterinary Medicine, Nicolaus Copernicus University in Toruń, 87-100 Toruń, Poland; dbukowska@umk.pl; 10Department of Veterinary Surgery, Institute of Veterinary Medicine, Nicolaus Copernicus University in Toruń, 87-100 Toruń, Poland; 11Bartosz Kempisty, Department of Histology and Embryology, Department of Anatomy, Poznań University of Medical Sciences, 6 Święcickiego St., 60-781 Poznań, Poland

**Keywords:** cardiomyocyte structure, cytoskeleton organization, extracellular matrix, cell culture, transcriptomic analysis

## Abstract

Numerous cardiovascular diseases (CVD) eventually lead to severe myocardial dysfunction, which is the most common cause of death worldwide. A better understanding of underlying molecular mechanisms of cardiovascular pathologies seems to be crucial to develop effective therapeutic options. Therefore, a worthwhile endeavor is a detailed molecular characterization of cells extracted from the myocardium. A transcriptomic profile of atrial cardiomyocytes during long-term primary cell culture revealed the expression patterns depending on the duration of the culture and the heart segment of origin (right atrial appendage and right atrium). Differentially expressed genes (DEGs) were classified as involved in ontological groups such as: “cellular component assembly”, “cellular component organization”, “cellular component biogenesis”, and “cytoskeleton organization”. Transcriptomic profiling allowed us to indicate the increased expression of *COL5A2*, *COL8A1*, and *COL12A1*, encoding different collagen subunits, pivotal in cardiac extracellular matrix (ECM) structure. Conversely, genes important for cellular architecture, such as *ABLIM1*, *TMOD1*, *XIRP1*, and *PHACTR1*, were downregulated during in vitro culture. The culture conditions may create a favorable environment for reconstruction of the ECM structures, whereas they may be suboptimal for expression of some pivotal transcripts responsible for the formation of intracellular structures.

## 1. Introduction

Many cardiovascular diseases (CVD) are associated with myocardial damage, and when untreated they eventually lead to severe myocardial dysfunction within a relatively short time. Clinically evident impairment of systolic myocardial performance predominantly results from the insufficient regeneration of cardiomyocytes (CMs) [1]. Currently available therapies are unable to replace lost CMs and largely irreversible cardiac dysfunction ensues. Therapeutic strategies aimed at stimulating the regenerative potential of myocardium may improve patient outcomes [2].

Strategies involving the transplantation of multipotent cardiac progenitor cells (CPCs) into the area of the injured myocardium, to promote regeneration of new functioning myocytes, are promising treatments. However, the effectiveness of the implantation strategy remains to be fully elucidated. Recent studies suggest that the adult heart is capable of limited cardiomyocyte turnover via the CPCs population [3,4,5,6]. Nevertheless, culture conditions and propagation of candidate cells need to be refined and standardized. A key developmental step will be to ensure in vitro culture conditions that allow cells extracted from the myocardium to expand. The description of the structure of the cultured cells will undoubtedly be an important element enabling the assessment of the condition of the cultured cardiomyocytes.

Understanding the structural property changes during myocardial damage may lead to the development of novel prevention therapies. Moreover, knowledge at the molecular level about the changes occurring in the participation of individual protein isoforms during either pathological processes or during in vitro culture will be crucial for determining the properties and usefulness of myocardial cells [7]. Therefore, novel measurement systems may lead to greater insight into cardiac tissue structure, properties, and performance [8]. Organization of the intracellular network cytoskeleton, or composition of the extracellular matrix (ECM), require deepening the knowledge at the molecular level to better understand the molecular changes in cells. The cytoskeleton tightly regulates myofibrillar activity and maintains muscle contraction/relaxation [9], while the ECM plays essential structural and regulatory roles in establishing and maintaining tissue architecture and cellular function [10].

The aim of the present study was to analyze the transcriptomic profile of genes involved in organization of the cardiomyocyte structure. Moreover, structural changes of cardiomyocytes isolated form the porcine right atrial appendage (RAA) and right atrium (RA) during long-term primary cell culture were evaluated to understand the CMs transcript expression patterns over the duration of the culture.

## 2. Materials and Methods

### 2.1. Animal Tissues

Pubertal crossbred Polish Landrace (PBZ × WBP) gilts (*Sus scrofa f. domestica*), bred with a mean age of 155 days (range 140–170 days) and a mean weight of 100 kg (95–120 kg), were used as a source of tissue. All animals were housed under identical conditions and fed the same forage. Porcine hearts were excised within 25 min of slaughter. After cutting the sternum and the diaphragm, the heart was removed. The hearts in intact pericardium were then preserved and transported to the laboratory on ice as soon as possible from a local slaughterhouse. After removing hearts from the pericardial sacs, the right atrial appendage and right atrial free wall were dissected and manually prepared with sterile surgical instruments to remove the visceral layer of the serous pericardium. Fat was also removed, and the tissue was the source of cells for in vitro culture. Each time, the delivered hearts were assessed for their suitability for downstream analyses. The following hearts were disqualified from further study: those with macroscopic injury after slaughter, those contaminated during transport or preliminary preparation, and those with connective tissue adhesions (signs of inflammation) or ischemia. As the research material is usually disposed of after slaughter, being a remnant by-product, no ethical committee approval was needed for the project.

### 2.2. Enzymatic Dissociation and Primary Cell Culture

The right atrial appendage (right auricle) and right atrium were extracted from the delivered hearts and washed in ice-cold PBS solution to remove the blood. After the two-step mincing by sterilized tools, the tissue underwent enzymatic digestion in DMEM + collagenase type II (2 mg/mL) solution at 37 °C for 40 min with gentle mixing. After the end of digestion, using nylon strainers of 70 μm pore size, the remaining tissue was separated from cell debris. The filtrate containing cells of interest was subject to centrifugation (5 min, 200× *g*, RT) to pellet the cells. The cells were washed with the PBS solution and then pre-plated on 25 mL culture bottles with culture medium (DMEM/F12, Sigma-Aldrich, Saint Louis, MO, USA), 20% FBS (Fetal Bovine Serum, Gibco, Thermo-Fischer Scientific, Waltham, MA, USA), 10% HS (Horse Serum, Gibco, Thermo-Fischer Scientific, Waltham, MA, USA), EGF (20 ng/mL; Sigma-Aldrich, Saint Louis, MO, USA), bFGF (10 ng/mL; Sigma-Aldrich, Saint Louis, MO, USA), and 1% P/S, and initially incubated for 4 h at 37 °C, 5% CO_2_. This stage aims to deplete the fibroblasts, which show much higher adhesion affinity [11,12]. After this stage, the supernatant (including nonadherent cardiomyocytes) was pelleted by centrifugation (5 min, 200× *g*, RT) and placed into the new 25 mL culture bottle, previously coated with 0.1% gelatin solution. The cells were cultured in DMEM/F12 medium complemented with 20% FBS, 10% HS, EGF (20 ng/mL), LIF (10 ng/mL), and 1% P/S. The procedure of cell isolation and culture is based on the current literature sources [11,12] modified to fit the aims of the experiments. The culture medium was changed every three days. Cultures selected for downstream analysis exhibited a cell viability of 90% and more, measured by the ADAM-MC Automated Cell Counter (Bulldog Bio Inc., Rochester, NY, USA) at all time intervals (7D, 15D, and 30D of culture).

### 2.3. Morphological Observation of Cells during Long-Term Primary In Vitro Culture

Using an inverted light microscope with relief contrast (IX73, Olympus, Tokyo, Japan), daily observation of cultured cells was performed. Images obtained during the culture period correspond to the time intervals used in the molecular analysis (7 days (D), 15D, and 30D).

### 2.4. Flow Cytometry Analysis

The cardiomyocytes levels were measured by flow cytometry. Cells were stained with combinations of the following antibodies: anti-α-Actinin (Sarcomeric)-FITC (cat.: 130-106-997), anti-Myosin Heavy Chain-APC (cat.: 130-106-253, Miltenyi Biotec, San Diego, CA, USA), and anti-GATA4 Alexa Fluor 488 (cat.: 560330, BD Biosciences, San Jose, CA, USA). Samples were fixed and permeabilized with ice-cold BD Perm/Wash™ buffer. For intracellular staining, to 100 μL of cell sediments, 300 μL of BD Perm/Wash™ buffer was added to each tube, mixed, and incubated on ice for 10 min. Following incubation, cells were centrifuged (1500 rpm, 5 min), supernatant was removed, and cell sediments were incubated on ice for 30 min with antibodies. Finally, cells were washed with BD Perm/Wash™ buffer. Stained samples were evaluated by flow cytometry using the BD FACSAria™ equipment (Becton Dickinson, Franklin Lakes, NJ, USA).

### 2.5. RNA Extraction and Reverse Transcription

RNA samples (both before and after in vitro culture) were isolated according to the Chomczyński and Sacchi [13] method, employing TRI reagent (Sigma-Aldrich; Merck KGaA, Saint Louis, MO, USA). Total RNA isolation was conducted using, successively, chloroform, 2-propanol, and 75% ethanol solution during the procedure. Next, the obtained RNA samples were re-suspended in 20–40 µL of RNase-free water and frozen at −80 °C. RNA integrity was determined by denaturing agarose gel (2%) electrophoresis. Subsequently, the total amount of the collected RNA was evaluated by measuring the optical density (OD) at 260 nm (NanoDrop spectrophotometer; Thermo Scientific, Inc., Waltham, MA, USA). RNA samples were subjected to reverse transcription using the RT2 First Strand kit (Qiagen, Hilden, Germany), according to the manufacturer’s protocol. To reverse-transcribe RNA into cDNA, 500 ng of an RNA sample was used [14].

### 2.6. Microarray Expression Study and Data Analysis

The microarray study was carried out as previously described [15]. Previously isolated total RNA (50 ng) from each pooled sample (during analyses, for each time interval we pooled RNA from 4 different cultures obtained from other hearts) was used in two rounds of sense cDNA amplification (Ambion^®^ WT Expression Kit). In the first step, synthesis of cRNA was performed by in vitro transcription (16 h, 40 °C). Subsequently, cRNA after purification was re-transcribed into cDNA. Then, cDNA samples obtained via the Affymetrix GeneChip^®^ WT Terminal Labeling and Hybridization kit (Affymetrix, Santa Clara, CA, USA) have been subjected to biotin labeling and fragmentation processes. Hybridization to the Affymetrix^®^ Porcine Gene 1.1 ST Array Strip was conducted at 48 °C for 20 h, in an AccuBlock™ Digital Dry Bath (Labnet International, Inc., NY, USA) hybridization oven. Our experiment employed 4 array strips. Two biological samples (each obtained by pooling RNA from four different animals to mitigate the effect of interindividual variation) were used for the analysis of each individual time interval. After hybridization, using an Affymetrix GeneAtlas™ Fluidics Station (Affymetrix, Santa Clara, CA, USA), all array strips were washed and stained, according to the technical protocol. The array strips were scanned using an Affymetrix GeneAtlas™ Imaging Station (Affymetrix, Santa Clara, CA, USA). The preliminary analysis of the scanned chips was performed using Affymetrix GeneAtlas TM Operating Software (Affymetrix, Santa Clara, CA, USA). The quality of gene expression data was checked according to quality control (QC) criteria following the manufacturer’s standards. The scans of the microarrays were saved as *.CEL files for downstream data analysis.

The created *.CEL files were subjected to further analysis performed using Bioconductor software, based on the R statistical language with the relevant Bioconductor libraries, as described previously [16,17]. To conduct background correction, normalization, and summarization of the results, we used the Robust Multiarray Averaging (RMA) algorithm. Assigned biological annotations were obtained from the “pd.porgene.1.1.st” library, employed for the mapping of normalized gene expression values with their symbols, gene names, and Entrez IDs, allowing us to generate a complex gene data table. To determine the statistical significance of the analyzed genes, moderated t-statistics from the empirical Bayes method were performed. The obtained *p*-values were corrected for multiple comparisons using Benjamini and Hochberg’s false discovery rate and described as adjusted *p*-values. The selection criteria of a significantly changed gene expression were based on an expression fold difference higher than abs. 2 and an adjusted *p*-value < 0.05. The list of differentially expressed genes (DEGs) was uploaded to the DAVID software (Database for Annotation, Visualization, and Integrated Discovery). Ontology groups that contained at least 5 genes and expressed a *p*-value (Benajamini) < 0.05 were selected for further analysis. Particularly, the “cellular component assembly”, “cellular component organization”, “cellular component biogenesis”, and “cytoskeleton organization” were selected as GO BPs of interest.

It is important to compare the expression profile in RA and RAA because the objective of the research was to characterize the molecular basis for structural changes. A Venn diagram was employed to detect relations between lists of DEGs in used hearts’ compartments and to explore the intersection of genes of analyzed terms from the functional analysis. Fifty of the most altered genes for both heart compartments were analyzed.

### 2.7. Real-Time Quantitative Polymerase Chain Reaction (RT-qPCR) Analysis

The RT-qPCR validation was performed using a Light Cycler^®^ 96 Real-Time PCR System, Roche Diagnostics GmbH (Mannheim, Germany), with SYBR Green as a detection dye. To the RNA material remaining after pooling of samples used in the microarray analysis (from 8 animals), 4 new biological replicates were added; as a result, the quantitative validation was performed for 12 samples for each time interval of culture. Levels of analyzed transcripts were standardized in each sample, in reference to hypoxanthine 1 phosphoribosyltransferase (*HPRT1*) and β-actin (*ACTB*) as an internal control. The final reaction mix consisted of 1 μL of cDNA, 5 μL of mastermix (RT² SYBR Green FAST Mastermix, Qiagen), 1 μL of forward + reverse primer mix (10 μM), and finally, 3 μL of PCR-grade water. We have used the Primer3 software for primer design (Table 1), based on Ensembl database transcript sequences. The exon–exon design method was used as an additional method to avoid potential remnant genomic DNA fragments’ amplification. For target cDNA quantification, we have performed relative quantification with the 2^−ΔΔCq^ method.

## 3. Results

Daily observation of cell morphology using an inverted microscope employing relief contrast (IX73, Olympus, Tokyo, Japan) was performed and documented (Figure 1 and Figure 2). Importantly, the morphology of cells obtained from different segments of the heart was similar. However, to obtain a complete view of the morphology of RAA and RA cells cultured in vitro, the information from the micrographs presented in two figures (Figure 1 and Figure 2) should be supplemented with previously published data [15]. The initially seeded cells exhibited an irregular, slightly elongated shape. After several days in culture (15D), the cells presented a clearly elongated, spindle-like morphology (Figure 2), resulting from the increase in confluency, until almost the entire available surface of the bottom of the culture bottle was covered at the end of the culture period (30D). Characteristically, cultured cells often tended to overlap and form three-dimensional structures, even in incomplete confluence (Figure 1, at 15D and 30D of culture). A similar observation was made regardless of the source of the cells (both RA and RAA cells showed a similar trend).

Flow cytometric analysis (Figure 3) revealed that both cells isolated from RAA and RA expressed cytoplasmic markers specific for CMs at all analyzed culture periods (at 7D, 15D, and 30D of culture). The expression of α-Actinin (Sarcomeric), Myosin Heavy Chain, and GATA4 was observed, and thus the obtained results confirm the effectiveness of a pre-plating step to select the CMs population [18,19,20]. The flow cytometry gating strategy is presented in Appendix A.

The dynamic changes in the transcriptome profile of cultured cardiomyocytes were observed at individual time intervals. Whole transcriptome profiling by Affymetrix microarray revealed gene expression after enzymatic dissociation and 7D, 15D, and 30D of cardiomyocytes long-term in vitro primary culture. Employing the Porcine Gene 1.1 ST Array Strip, microarray transcriptome screening was performed. The cells obtained from both RAA and RA at individual time intervals resulted in changes in gene expression levels. Genes with fold change >|2| and an adjusted *p*-value of <0.05 were considered as differentially expressed. Global expression analysis yielded 4239 DEGs for RA and 4662 DEGs for RAA. Principal component analysis (PCA) of overall DEGs allowed to examine the variance between the analyzed sample groups [15].

Genes related to the cellular structure organization during development and maturation were analyzed, revealing 237 different transcripts for RA and 276 for RAA. In the next step, using DAVID software, selected DEGs were classify as involved in ontological groups, such as: “cellular component assembly”, “cellular component organization”, “cellular component biogenesis”, and “cytoskeleton organization”. Genes selected by DAVID software analysis were subjected to hierarchical clustering and are presented as heatmaps (Figure 4). Additionally, comprehensive DEGs enrichment analysis was performed, and obtained results are presented in the form of a clustergram (Appendix A). We used the *Enrichr* database [21,22,23] for the determination of ontological groups. Selected DEGs were used as input genes for the analysis. The obtained enriched processes belonging to the GO Biological Process 2021 group were ranked according to *p*-value. The names and GO signatures of the ten most enriched processes are listed in a bar graph. The same ten processes were used to prepare the clustergram. The most enriched processes are related to the ECM (extracellular matrix organization GO:0030198, extracellular structure organization GO:0043062, external encapsulating structure organization GO:0045229, and supramolecular fiber organization GO:0097435).

Due to the structure of the GO database and since any given gene can belong to multiple GO groups, gene intersections between selected ontological terms were examined. The relationships between genes and GO terms were mapped with circle plots, with visualization of logFC values and gene symbols (Figure 5). All the genes were either upregulated or downregulated in the cell culture intervals compared to the inoculation.

The multitude of results obtained for cells from two separate cardiac segments contributed to the creation of a Venn diagram visualizing the gene expression patterns’ comparison (Figure 6). Out of differentially expressed genes (DEGs) sets for RA and RAA containing genes belonging to the GO BP terms of interest, the 50 most altered genes for both segments were selected and presented on the diagram. Commonly upregulated and commonly downregulated genes in RA and RAA are visible in the middle part of the diagram (Figure 6). Moreover, the same direction of changes in the transcript levels can be observed for RA and RAA. In the lateral parts of the Venn diagram, the gene names listed indicate the DEGs specific only to RAA (right side) and RA (left side of the diagram).

High-throughput global transcript analysis employing the microarray approach is largely qualitative. Therefore, the RT-qPCR method was applied. RT-qPCR analysis together with microarray results are illustrated as a heatmap (Figure 7). As can be seen, the RT-qPCR analysis confirmed the direction of expression change for all the selected DEGs. Importantly, none of the analyzed genes showed any other direction of expression changes than those indicated by the results of the expression microarrays. However, the scale of differences in transcript levels differed between both methods analyzed. The most upregulated genes, in RT-qPCR, from the examined DEGs included, among others, *TNC* (tenascin-C), *ITGA8* (integrin, α 8), and *COL12A1* (collagen type XII α 1 chain). The strongest downregulated genes were *ABLIM1* (actin-binding LIM protein 1), *DMD* (dystrophin), and *MATN2* (matrilin 2).

## 4. Discussion

It is necessary to understand the mechanisms regulating cardiac muscle cells that may enable application in cell-based therapy. The myocardium could serve as a source of cells that could be expanded as a treatment platform [5,24]. Therefore, the detailed characterization of cells obtained from the myocardium is important. Cardiomyocytes have been demonstrated to exhibit the ability to develop new vessels [15]. Furthermore, epigenetic alterations by measurement of mRNA expression levels of the DNA methyltransferases during CMs in vitro culture were evaluated in the current study [25]. The reported culture conditions may create a favorable environment for neovascularization; nevertheless, expression levels of genes encoding structural proteins, such as myosin (*MYH7*, *MYL3*) or actin (*ACTC1*), were significantly lower during culture. The previous results and the fact that the correct structure and properties of cardiomyocytes are necessary for the overall systolic/diastolic activity of this essential organ directed us to focus on issues related to the analysis of the molecular background of key factors for the correct structure of the growing cultured cells. Porcine hearts were chosen for this study because they are important due to their many similarities to human beings. Moreover, large animal models are essential to develop the discoveries from murine models into clinical therapies and interventions.

The extracellular matrix (ECM) is the non-cellular, highly dynamic structural network composed of numerous glycoproteins, such as collagens, elastin, fibronectin, and laminins [26]. ECM remodeling is crucial for regulating the cellular behavior, development, and maturation [27]. The ECM provides structural support for the developing myocardium and plays a crucial role in cardiac homeostasis by transducing key signals to cardiomyocytes or vascular cells [28]. Various natural components (such as collagen [29] or Matrigel [30]) and synthetic [31] biomaterials have been used to increase the efficiency of cardiomyocyte culture and influence the degree of maturation obtained. However, none of these biomaterials can fully recapitulate the architecture and functional composition of the cardiac ECM. It has been reported that human pluripotent stem cell-derived cardiomyocytes (hPSC-CMs) obtain the most favorable conditions for growth and maturation when used in artificial ECM [32] or decellularized myocardial ECM [33,34,35]. Therefore, any changes in the structure and composition of the ECM require analysis as it is crucial for the proper functioning and maturation of cultured cells.

The cardiac ECM is primarily composed of fibrillar collagen type I, which is commonly found in tissues’ ECM together with type V collagen, which helps to organize type I collagen and makes contact with basement membranes [36], and the less abundant collagen type XII [37]. During long-term in vitro culture, cells isolated from both the right atrial appendage (RAA) and right atrial (RA) wall exhibited increased expression of three genes, *COL5A2*, *COL8A1*, and *COL12A1*, encoding different collagen subunits. Interestingly, increased transcript levels of these genes were observed as the culture continued in RAA cells. The obtained results on the mRNA level may suggest the readiness of cells maintained in the conditions of long-term in vitro culture to reconstruct the ECM structures precisely by the increased production of the analyzed transcripts. The transcript level that is translated into protein under 2D culture conditions would indicate the extent that this structure can be reconstructed by collagen overproduction. Azuaje et al. showed that *Col5a2* transcript is highly expressed in the left ventricle after myocardial infarction (MI), suggesting a role of collagen V α 2 chain in the post-MI response via ventricular remodeling by recruiting and deposition of sufficient amounts of collagen type I, thereby increasing myocardium healing [38]. In contrast, transgenic mice expressing a non-functional form of *Col5a2* do not present ventricular defects [39]. Moreover, patients suffering from classic Ehlers–Danlos syndrome, a rare connective tissue disorder mainly caused by mutations in *COL5A1* or *COL5A2*, do not appear to show ventricular malformations [40]. During transcriptomic analyses on a mouse model of cardiac remodeling, Wang et al. reported upregulation of transcript levels of the gene encoding collagen type VIII α 1 chain [41]. The authors suggest that the *Col8a1* gene may be a candidate for one of the biomarkers for cardiac remodeling. Interesting results presented in studies comparing mRNA expression levels of extracellular matrix protein genes with functional changes in the heart that occur with reproductive status indicated that *Col8a1* mRNA levels increased in the early postpartum period [42]. Other investigators showed the effect of deletion of the *Col8a1/2* genes in mice on induced pressure overload of the heart and reduced fibrosis, but increased dilatation and mortality compared to wildtype controls [43]. Therefore, different collagens show a significant role in the process of restoring matrix structures.

Among the differentially expressed genes identified in the present study, two genes encoding integrins (*ITGA8* and *ITGB6*) indicated significant changes in the mRNA levels during culture in relation to the starting point of in vitro culture. Integrins are heterodimeric, transmembrane glycoprotein receptors important for the ECM, and thus the cells, structure, and signaling. Members of this family are capable of bidirectionally transmitting signals between the ECM and the intracellular environment [44]. The obtained results are not as unequivocal as in the case of collagen-encoding genes, because *ITGA8* showed increased mRNA expression levels, while *ITGB6* showed downregulation. Liu et al. suggested an important role of *Itga8* expression in the molecular mechanism of hypoplastic left heart syndrome (HLHS), based on the results of a mouse model [45]. Employing the next-generation high-throughput RNA sequencing, Chen et al. showed, in the neonatal mouse heart, that Itgb6 belongs to targets of highly dysregulated lncRNAs during the first seven days of cardiac development [46]. In cells isolated from both the right atrial appendage (RAA) and the right atrial (RA) wall during long-term culture, the highest level of increased transcript expression was observed for genes encoding another extracellular matrix protein: tenascin-C (TNC). It is important to note that tenascin-C is abundantly expressed in the heart during embryonic development but is rarely detected in healthy subjects [47]. Nevertheless, TNC is a very important biomarker for many pathological conditions associated with inflammation, such as myocardial infarction, hypertensive cardiac fibrosis, or heart failure with preserved ejection fraction [47,48,49]. These observations suggest that tenascin-C is re-expressed in the affected heart. The in vitro culture conditions shock the cultured heart muscle cells by radically changing the environment in which they must exist. Therefore, another of the mechanisms of cell adaptation, aimed at restoring optimal conditions for development, is a significant overproduction of the *TNC* transcript.

The pivotal role for ECM homeostasis and its composition dynamic remodeling has been confirmed for matrix metalloproteinases (MMPs), which are an important environmental mediator of cardiac diastolic or systolic dysfunction [50]. Members of the matrix metalloproteinase (*MMP*) gene family, and encoded by these genes’ proteins, that are zinc-dependent proteases, are involved in the breakdown of ECM [51]. MMP2 and MMP9 belong to the most frequently analyzed MMPs in relation to HF syndrome and other cardiovascular diseases [52,53]. Interestingly, the screening analyses of the transcriptome showed overexpression for both *MMP2* and *MMP9* at all time intervals of the culture. It should be emphasized that the increased levels of *MMP2* mRNA were only observed in RA, while the overexpression of the *MMP9* gene was demonstrated in cells obtained from RAA. Hayashidani et al. reported the response of MMP2 KO mice to in vivo myocardial infraction (MI) and described decreased mortality rates in the week following MI [54]. Moreover, the authors found protective effects in terms of hemodynamic performance or remodeling [54]. Other studies [55] employing a swine model, which as a large animal model may be more clinically representative than the most commonly used small rodent models, also confirmed the role of MMP in heart disease progression. Apple et al., using the MMP inhibitor (1 mg/kg PGE-530742) and starting animal treatment 3 days prior to MI, showed decreased left ventricular (LV) dilation post-myocardial infarction and improved LV contractile performance [55]. Numerous scientific reports confirm that MMP2 and MMP9 have been recognized to play an important role in matrix remodeling in these cardiac disease states [52]. However, results presented by Kiczak et al. indicated no differences in the expression of transcripts and proteins of MMP9 and MMP2 in either LV myocardium or skeletal muscles between diseased and healthy animals. These data may suggest that the activity of these enzymes may be altered post-translationally in HF and is not dependent on the expression of the mRNA or protein [53]. It should be emphasized that the MMPs are associated with the state of active myocardial remodeling and could be a potentially useful marker for the identification of patients at risk for heart failure development and poor outcome.

Variable expression levels of transcripts for genes encoding key structural proteins were also observed. Myosin, the major component of the thick filaments, is an essential motor protein of the myocardium. Lower mRNA levels of *MYH7* and *MYL3*, genes encoding two isoforms of myosin heavy and light chains, were previously observed [15]. In all culture intervals, both in RA and RAA cells, upregulation of *MYO5A* transcript levels was observed. The myosin Va class is involved in the cytoplasmic transport of vesicles along actin filaments to membrane docking sites [56]. MYO5A is a member of the myosin V heavy-chain class of actin-based motor proteins and may play an important role in potassium channel functioning via modulation of trafficking pathways and cell surface density of Kv1.5 in the myocardium [57]. A change in the expression levels of several genes encoding proteins closely related to the actin structure was observed. In contrast to an upregulation of *MYO5A* mRNA levels, *ABLIM1*, *TMOD1*, *XIRP1*, and *PHACTR1* were downregulated during in vitro culture. Regulation of actin filament organization and dynamics is important for cellular architecture and numerous biological functions, including muscle contraction. Actin-binding LIM 1 (abLIM1, gene: *ABLIM1*) is a cytoskeletal protein that has been implicated in interactions between actin filaments and cytoplasmic targets [58]. Additionally, this actin-binding intracellular protein is localized to the sarcomeric Z-disk of the myocardium, playing a role in force transmission and muscle integrity [59]. Tropomodulin1 (Tmod1, gene: *TMOD1*) is an actin-capping protein necessary in actin thin-filament pointed-end dynamics and length in striated muscle, including cardiac muscle [60]. Bliss et al. demonstrated that in cardiac myocytes, the length regulatory function is modulated by the Tmod1 phosphorylation state [61]. Filamins (FLNs) are large dimeric actin-binding proteins, and as scaffolding proteins, FLNs regulate actin cytoskeleton remodeling [62]. Employing transcriptomic analyses, we indicated increasing levels of *FLNB*, encoding filamin B isoform. Interestingly, Xin actin-binding repeat-containing protein (gene: *XIRP1*), an important filamins ligand affecting the behavior of FLNs family members in Z-discs and myotendinous junctions of striated muscle cells [63], showed significantly decreased expression of the transcripts under the provided culture conditions. Leber et al. emphasized the role and correlation between Xin actin-binding repeat-containing protein and filamin family members in myofibrillar myopathy and skeletal and cardiac diseases [64]. Finally, it was demonstrated that lower mRNA levels of the phosphatase and actin regulating protein 1 were encoded by the *PHACTR1* gene during cell culture. *PHACTR1* transcript expression was originally observed in brain tissue [65], whereas subsequent studies indicated the presence of different variants of transcripts also in other types of tissues, including heart and coronary vessels [66]. The PHACTR protein family interacts directly with actin and protein phosphatase 1 (PP1) [65]. By binding to PP1, PHACTR1 acts as an inhibitor of its activity. Other results indicated a role of PHACTR1 in altering the structure of actin in HUVECs cell lines, where a reduction of F-actin filament numbers and repartitioning as well as increasing cell protrusion dynamics were observed [67]. The authors suggest that through its correlation with VEGF, PHACTR1 is a key component in the angiogenic process [67]. PHACTR1 function is linked with atherosclerosis-relevant phenotypes, such as angiogenesis [68], extracellular matrix protein production [69], and inflammation [70] in vitro. Numerous scientific reports emphasize the connection of *PHACTR1* transcript expression with coronary artery disease (CAD) initiation and progression [66,71,72,73].

Other proteins crucial for the structure of the heart tissue, whose transcript levels decreased significantly under in vitro culture conditions, are dystrophin (gene: *DMD*) and calsequestrin 2 (gene: *CASQ2*). The protein encoded by the *DMD* gene forms a component of the dystrophin-associated glycoprotein complex (DGC), which bridges the inner cytoskeleton and the ECM [74]. Heart muscles lacking functional dystrophin are mechanically weak, and contraction of cardiac myocytes leads to membrane damage, which affects the calcium channels, resulting in increased levels of intracellular calcium and activating proteases that consequently degrade contractile proteins, promoting cellular death and fibrosis [75]. Calcium handling in heart tissue belongs to the roles of calsequestrin family members. CASQ2 is localized to the sarcoplasmic reticulum (SR) in myocardium and functions as a calcium storage protein [76]. Refaat et al. showed a correlation between *CASQ2* gene polymorphism and sudden cardiac arrest (SCA) due to ventricular arrhythmias (VA) in patients with CAD [77].

The final aim of our project was to develop a strong background to understand the in vitro situation of porcine cardiomyocytes as a mode that will eventually be applicable in clinical practice. Atrial appendices, either left or right, may be harvested during minimally invasive procedures without any consequences. Moreover, during routine cardiac procedures employed in cardio-pulmonary bypass (all standard cardiac surgical procedures on the cardiac valves, aorta, heart transplantations, and many congenital malformations), the right atrial appendage is usually removed (the site of choice to introduce venous canula). The volume of the removed atrium can reach even a few square centimeters. Contrary to the atrium, in daily cardiac surgical practice, it is not possible to harvest such an enormously large volume from the left or right ventricular myocardium. The biggest biopsies are taken routinely from the right aspect of the ventricular septum in patients after heart transplantation to check for rejection (endomyocardial biopsies). However, the volume of the largest of them is approximately 1 mm^3^. Therefore, the best candidate source of cells is the atrial myocardium.

The transcriptomic profile of genes encoding important structural proteins clearly demonstrates that long-term 2D culture conditions do not sufficiently allow for the reconstruction of important elements of the cytoskeleton. Moreover, further analysis of the observed changes at the protein level will provide more complete information about changes occurring during the conducted long-term primary cell culture. The present report is an initial step in developing a novel strategy to understand the in vitro situation of porcine cardiomyocytes. Our goal was to better understand the molecular background of the physiological potential of myocardial regeneration in an animal model of healthy hearts. The initiated research is required to be continued through, among others, determination of proteins for which changes at the transcript level have been considered essential for structural changes in cells. Similarly, the implementation of analyses using 3D cultures as well as functional analyses should be the next step to characterize CMs under long-term in vitro culture conditions. Significantly decreased mRNA expression levels indicate a potential problem with the lack of specific “building blocks” for intracellular structures, and thus cardiomyocytes’ development and final maturation may be limited.

## 5. Conclusions

The effect of long-term primary cell culture conditions on the expression levels of gene transcripts encoding important structural proteins demonstrates increasing mRNA expression levels of genes encoding collagen subunits (*COL5A2*, *COL8A1*, and *COL12A1*) or tenascin-C (*TNC*), which may create a favorable environment for reconstruction of the ECM structures. A significant limit of the applied 2D in vitro culture conditions is the significantly lower expression of some pivotal transcripts responsible for the formation of intracellular structures. Decreased levels of genes such as *DMD*, *CASQ2*, *ABLIM1*, *TMOD1*, *XIRP1*, and *PHACTR1* confirm the necessity for further development and optimization of cardiomyocyte culture conditions. The conclusions drawn from the results are an important cognitive step in revealing the therapeutic potential of the cells.

## Figures and Tables

**Figure 1 genes-13-01205-f001:**
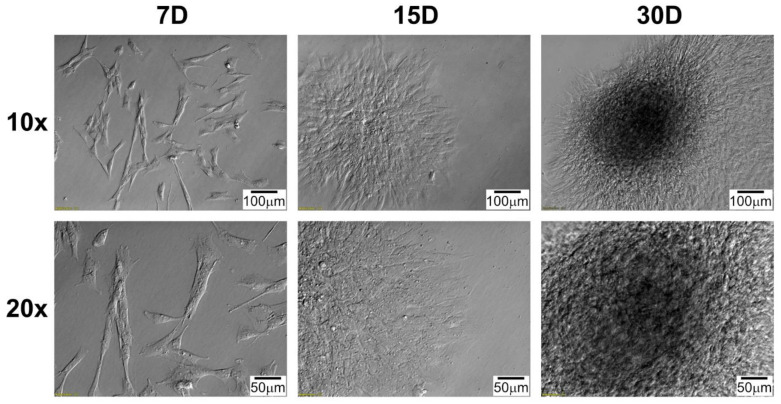
Changes in right atrial appendage (RAA) cells’ morphology during long-term in vitro primary culture at individual time intervals. D: day of culture; 10×, 20×: magnification.

**Figure 2 genes-13-01205-f002:**
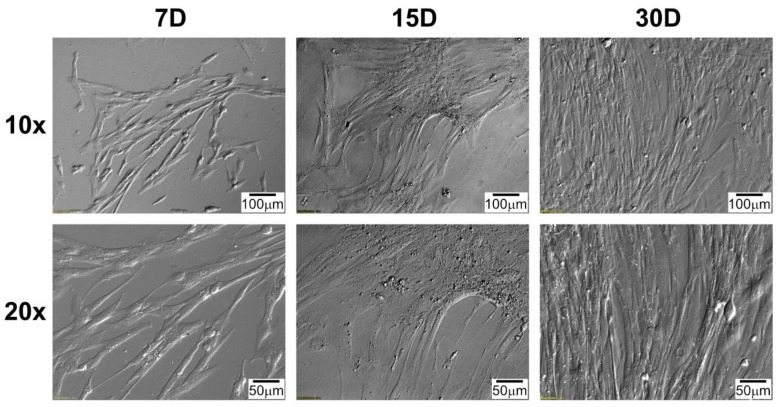
Changes in right atrium (RA) cells’ morphology during long-term in vitro primary culture at individual time intervals. D: day of culture; 10×, 20×: magnification.

**Figure 3 genes-13-01205-f003:**
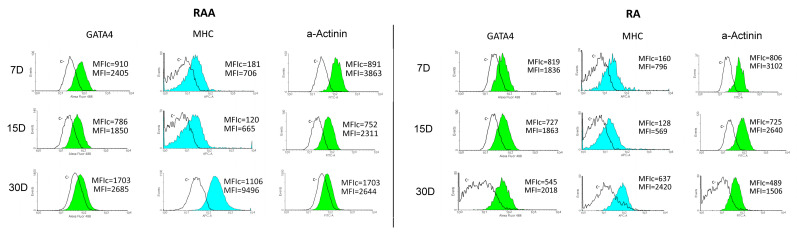
The results of flow cytometry analysis of selected cardiomyocytes markers (α-Actinin (Sarcomeric), Myosin Heavy Chainm and GATA4) in the cell samples at different time intervals subjected to in vitro culture. D: day of culture; MFI: mean fluorescent intensity; MHC: Myosin Heavy Chain.

**Figure 4 genes-13-01205-f004:**
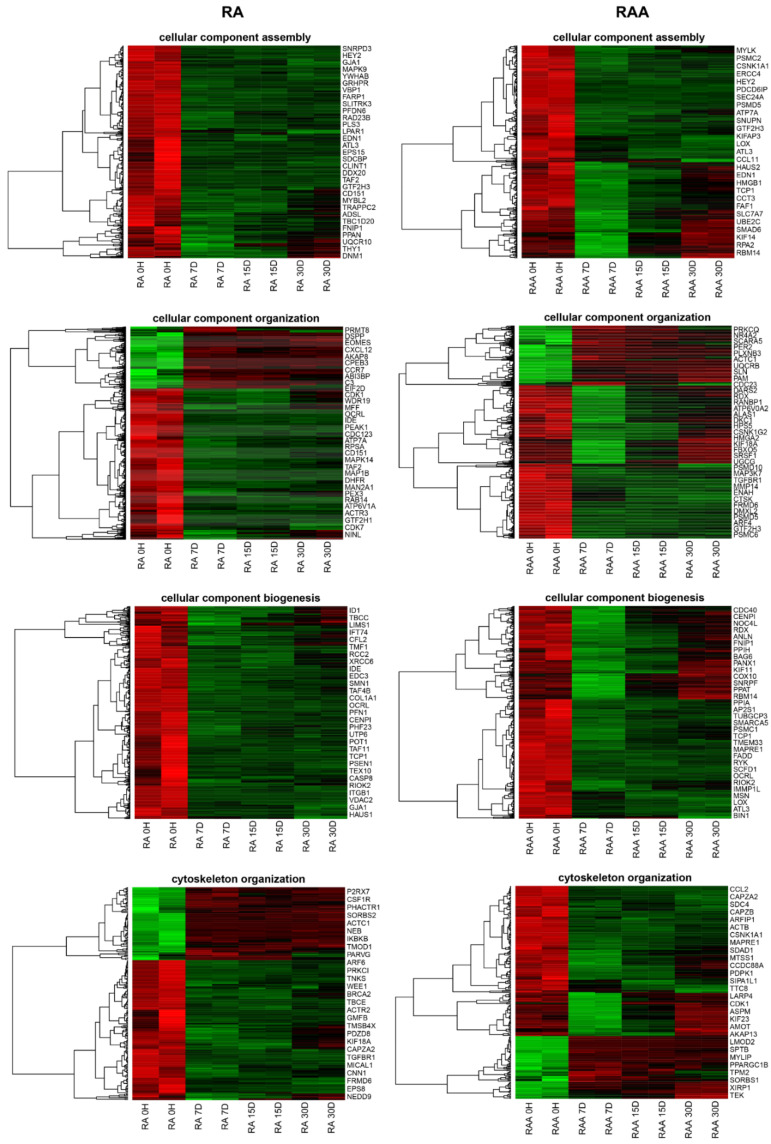
Heatmaps with hierarchical clustering of the differentially expressed genes, both in the right atrial appendage (RAA) and right atrium (RA), involved in “cellular component assembly”, “cellular component organization”, “cellular component biogenesis”, and “cytoskeleton organization”, based on GO BP terms. Each separate row on the y-axis represents a single transcript. The red color indicates downregulated, and the green indicates upregulated genes at subsequent time intervals of the analysis.

**Figure 5 genes-13-01205-f005:**
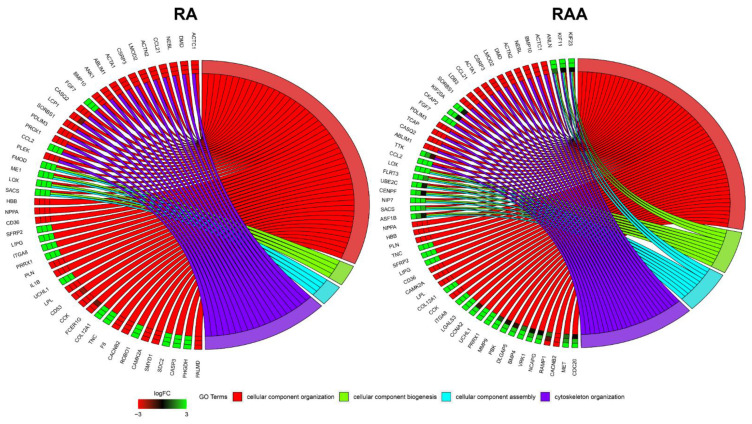
Analysis of enriched Gene Ontology groups involved in general cellular structure organization in cultured porcine cardiac muscle cells. The ribbons show the genes belonging to the given categories. The color bars near each gene correspond to logFC between culture intervals.

**Figure 6 genes-13-01205-f006:**
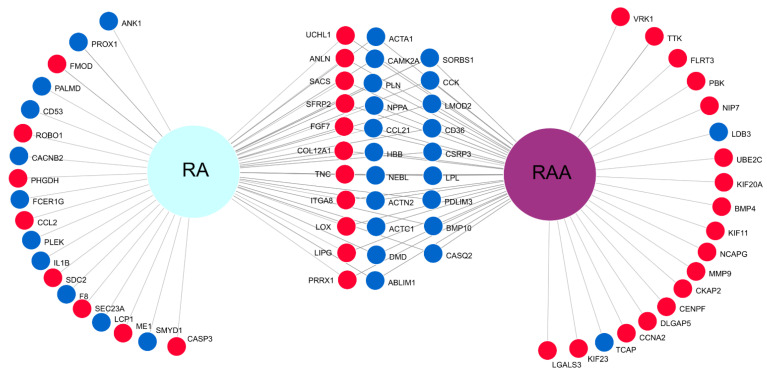
Venn diagram. The 50 most altered genes from analyzed GO BP terms for RAA and RA were used in the comparison. Blue circles show downregulated genes, whereas red circles show upregulated.

**Figure 7 genes-13-01205-f007:**
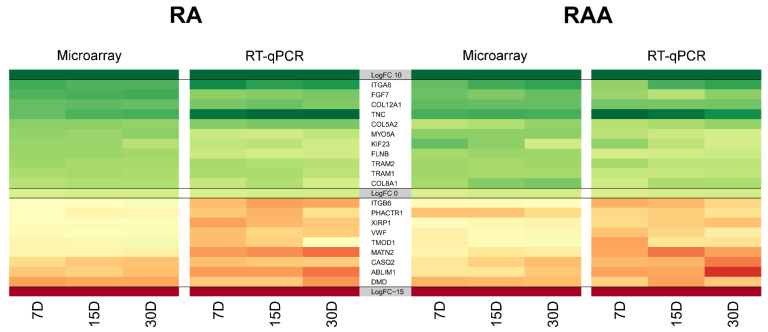
Heatmap representation of selected differentially expressed genes. Microarray results are presented together with RT-qPCR quantitative validation to indicate common patterns of transcript expression. Quantitative data (LogFC), used to compile the figure, are available in Appendix A. All the presented sample means were deemed to be statistically significant (*p* < 0.05). A scale indicating the color and its intensity depending on the direction and size of the changes has been integrated into the figure.

**Table 1 genes-13-01205-t001:** Primers. Oligonucleotide sequences of primers used for RT-qPCR analysis.

Gene	Primer Sequence (5′-3′)		Product Size (bp)
*ITGA8*	F	CCAGCAGACCAAAACCCTTC	164
	R	AAGAAGTTGTGCAGCTGTGG	
*TNC*	F	TTTCAGATGCCACCCCAGAT	169
	R	GTGGCTTCTCTGAGACCTGT	
*FGF7*	F	TGGAAATCAGGACAGTGGCT	192
	R	CTCCTCCACTGTGTGTCCAT	
*COL12A1*	F	TCCACAGGTTCAAGAGGTCC	150
	R	TTGTTAGCCGGAACCTGGAT	
*KIF23*	F	TGCAACAGGAGCTTGAAACC	243
	R	AGGGTCTCTCTGGCTTTTCA	
*FLNB*	F	AACATCCCGAACAGCCCTTA	159
	R	ACTGACATCACCTTCCCCAG	
*COL5A2*	F	TGGTGAAAATGGCCCAACTG	193
	R	TCCTCGACCACCTTTCAGTC	
*COL8A1*	F	GGAGAGAAGGGCTTTGGGAT	249
	R	GATCCCATCCTGACCTGGTT	
*MYO5A*	F	TGAGAAGAAGGTGCCTCTGG	199
	R	TTCCTGACGCTTGAGTGACT	
*TRAM1*	F	CCTCGTCAGCTCGTCTACAT	241
	R	AGCCAACAGTGAGTACCGAA	
*TRAM2*	F	ACATCTGCCTGTACCTGGTC	209
	R	GGCGAGAGTGAGGATGAAGA	
*DMD*	F	TCCACTTCTGTCCAAGGTCC	187
	R	GCAGTCTTCGGAGCTTCATG	
*ABLIM1*	F	ATGAAGCTCAACTCAGGCCT	161
	R	TAGCCTGGGAGAGATGAGGT	
*CASQ2*	F	TCCTTGTCTATGCAACGGGT	187
	R	GCTTTTCCCAGGTGTTGAGG	
*TMOD1*	F	ACAGCCGGGTCATAGATCAG	159
	R	GTCAGGGTCCAACTCATCCA	
*XIRP1*	F	AGAGCAATGCAGTGAGGACT	201
	R	AGTCCTTCTCGTCCACCAAG	
*PHACTR1*	F	TTAACTCGGAAGCTCAGCCA	174
	R	GAGCTCCTTTCGAATGGCAG	
*MATN2*	F	ACGACTTGCAGAATCCAGGA	156
	R	TGAGGCACAGTAGTCCACAG	
*VWF*	F	TGCAACACTTGTGTCTGTCG	229
	R	TGCATTTCAGGGAGGGGTAG	
*ITGB6*	F	TGACGACCTCAACACGATCA	190
	R	TCCAAAGGTAGGCAAGCAGA	
ACTB	F	CCCTGGAGAAGAGCTACGAG	156
	R	CGTCGCACTTCATGATGGAG	
HPRT1	F	CCATCACATCGTAGCCCTCT	166
	R	TATATCGCCCGTTGACTGGT	

## Data Availability

Not applicable.

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
