# Peer review of "Transcriptomic Profile of Genes Regulating the Structural Organization of Porcine Atrial Cardiomyocytes during Primary In Vitro Culture"

_genes, 2022, doi:10.3390/genes13071205_

Round 1
Reviewer 1 Report
In the present article, Nawrocki and co-workers depickted the transcriptomic profile of porcine primary culture of atrial cardiomyocytes. Gene expression levels were monitored at different culture periods up to one month and differentially expressed genes were clustered by ontology into four main groups most of them involved in the cellular structure and ECM maintenance. The authors concluded that the long term culture condition might increase the expression levels of ECM genes which might be related to long term cardiomyocytes ECM remodeling. As a whole the manuscript is well written and the presented data sounds interesting. However some few points still need to be adressed:
Major comments:
1- Line 215, Figures 1 and 2: Unlike what was promoted in the manuscript, cardiomyocytes from the RA and RAA do not present the same morphological aspects nor relatively similar sizes. This is more evident at 30D of culture. Please provide an explanation for this difference or present a more representative figure.
2- Figure 3: The resoltion of the figure is very low that it is bearly readible. Please provide a better resolution panel.
3- Line 120: "Cultures selected for downstream analysis exhibited a cell viability of 90% and more measured by ADAM-MC Automated Cell Counter": Did the authors mean 90% of viability after 30 days of in vitro culture? please specify.
Minor comments:
Few spelling/ sentence structres need to be reviewed such as:
Line 72: Too long sentence that the meaning is lamost lost. Please reformulate.
Line 317: repeated "culture conditions", please ommit the repetition.
Reviewer 2 Report
This study reveals new gene markers regulating the structural organization of porcine atrial cardiomyocytes during primary in vitro culture, hence is interesting.
However, in the Introduction or Discussion, the reason for selection of porcine atrial cardiomyocytes rather than other animals' should be given.
Besides, in reference 77, the journal name is 'Heart Rhythm' rather than 'Hear. Rhythm'.
Reviewer 3 Report
The article presents observations leading to the conclusion that long-term primary cell culture can influence cellular gene expressions. This can be a confounding factor while evaluating the impact of a pathological condition and hence bears applied importance.
1. Title of the study can be improved as it seems defocused on the key observation.
2. The differential expression of the genes during the long-term primary culture may not truly represent the in vivo conditions. In addition, the study design is not sufficient to make a comment on the differential expression of genes involved in the structural organization of atrial cardiomyocytes in vivo. This seems an over-interpretation.
3. Authors could discuss better the applied significance of their findings. They should also comment on whether similar findings have been reported before in the literature, if yes, then, that should be discussed in detail. Accordingly, the conclusion of the study should be revised.
4. Most of the discussion has been wasted on describing the ontological characteristics and the known pathological involvement of the differentially regulated genes, this is of not much use for the readers and can be easily accommodated in a table.
5. In multiple places, it's hard to understand what authors wanted to convey, a reframing of the sentences and grammatical corrections can improve the meaning. A thorough language editing is required.
